# New data on the validity of the Fazio Laterality Inventory

**Wojciech Łukasz Dragan** [1]☯*, **Andrzej Śliwerski** [2]☯, **Monika Folkierska-Żukowska**[1]

**1** Faculty of Psychology, University of Warsaw, Warsaw, Poland, **2** Institute of Psychology, University of Łódź, Łódź, Poland

☯ These authors contributed equally to this work.
\* wdragan@psych.uw.edu.pl

**Data Availability Statement:** All relevant data are within the manuscript and its Supporting information files.

**Funding:** This study was funded by National Science Centre Grant No. 2014/15/B/HS6/03754

## Abstract

The Fazio Laterality Inventory (FLI) is a recent measure of handedness. Although initially validated, there is still a lack of studies assessing its psychometric properties in samples outside the USA. The present study explores the validity of the Polish adaptation of the FLI. We used data gathered from a convenience sample of 727 participants. They completed the FLI and the Edinburgh Handedness Inventory to establish concurrent validity. Confirmatory factor analysis was used to investigate the factor structure of the FLI. In addition, an Item Response Theory (IRT) model for continuous item scores was also used to identify the discrimination and difficulty parameters of the FLI items. The Polish version of the FLI was characterized by good reliability indices and has high concurrent validity with the Edinburgh Handedness Inventory. We identified a bi-factorial structure for the questionnaire. The IRT analyses showed that the FLI items have good discrimination and difficulty parameters. Our study provides new insights into the properties of the Fazio Laterality Inventory.

## 1. Introduction

Left–right asymmetry is a phenomenon related to differences in the orientation of visceral organs across the mediolateral plane in animals. The consequences of this feature are also visible in the structure, circuitry, and function of the brain [1]. Brain asymmetries are linked to a number of functional asymmetries in behavior [2]. One such asymmetry in humans is laterality—a preference for one side of the body.

Its most obvious manifestation is handedness. Approximately 90% of the population is biased toward using their right hand in everyday activities [3]. Recording this preference using a questionnaire is the standard method of assessing handedness in psychological literature [4], and many handedness questionnaires have been developed so far. The Edinburgh Handedness Inventory (EHI) is most commonly used [5]. Fazio, Coenen & Denney [6] pointed out several shortcomings of the EHI (e.g., outdated items; instructions being difficult to understand for respondents with lower education; a response format that facilitates an extreme response style). Due to these problems, some alternative measures of handedness have been proposed recently, such as the Flanders Questionnaire [7], the Home Handedness Questionnaire [8], and the Fazio Laterality Inventory (FLI) [9].

and internal funding from the University of Warsaw.

**Competing interests:** The authors have declared that no competing interests exist.

The FLI was introduced as a modernized measure of handedness by Fazio, et al. [9]. It is composed of ten items regarding manual preference in different domains of movement, such as fine motor skill, ballistic movement, social/communicative movement, and movement involving the midline of the body. Fazio, et al. [9] demonstrated that the FLI has good psychometric properties. Their analysis revealed that all items of the FLI loaded on one single factor. In a further validity study by Fazio & Cantor [10] on a population with established atypical handedness, a two-factor solution was found. The first factor includes items related to fine motor/ballistic movements and the second one comprises items related to expressive/instrumental movements. The FLI is still a new measure with limited data on its validity (including its factor structure). For this reason, we undertook a study to further validate the FLI with the use of an independent sample. The goals were to verify the basic psychometric properties of this questionnaire (including reliability and its concurrent and factorial validity). Concurrent validity was identified by analyzing correlations of the FLI with the Edinburgh Handedness Inventory. While examining the factorial structure of the FLI we employed confirmatory factor analysis. We considered different factor solutions, including those which were identified by Fazio and colleagues (2013, 2015), as well as second-order and bi-factor solutions which are present in handedness literature [11, 12]. The latter seem to be the most appropriate choice due to the multidimensionality of the FLI identified by Fazio & Cantor [10]. What is important here is that this characteristic arises when responding to measuring a broad construct of handedness. In such cases, Reise [13] suggests implementing the bi-factor model for describing the factor structure of the measure.

We also broadened our study on the psychometric properties of the FLI by using the Item Response Theory (IRT) method. IRT models help explain the relationship between latent traits (symbolized by Ɵ) and their manifestation in questionnaire responses—in simple terms, they help determine the precision of psychological scales. It is said that the stronger the trait is, the higher is the probability of a correct response (i.e., that the answer follows the key). However, IRT models assume that the items of a scale are not equally informative across the latent trait range. Their informative value depends on two item parameters: difficulty and discrimination. Difficulty parameters indicate the threshold of a latent trait between answer choices and the discrimination parameter reflects the degree to which an item discriminates individuals across the latent trait range [14].

Samejima [15, 16] proposed an IRT model for continuous item scores as an extension of her graded response model. Unlike the popular binary and polytomous IRT models, the continuous response model (CRM) is theoretically more correct but less simple [17]. CRM parameters are estimated using the marginal maximum likelihood and Expectation-Maximization (EM) algorithm proposed by Shojima [18]. Discrimination and difficulty parameters in this model have practical meaning and are interpreted in the same manner as in the binary and polytomous IRT models [19].

To date, there have been only a few attempts to apply IRT models to laterality data, and these have mostly been to estimate construct validity. For example, Dragovic [20] examined the dimensionality of the EHI using latent class analysis. The same method was used by Dragovic & Hammond [21] to identify the structure of the Annett Hand Preference Questionnaire (AHPQ). Thus, the analyses presented here are the first attempt to identify difficulty and discrimination indices of items of a laterality measure using an IRT approach.

## 2. Materials and methods

### 2.1 Participants

A total of 727 people volunteered to participate in the study. After the removal of missing data, 604 people (410 women) took part in the study. Participants were recruited in Poland via an

online survey that was distributed through Facebook (organic traffic), leaflets, posters, and word of mouth. Demographic characteristics are presented in Table 1.

The study was approved by the Ethics Board of the Faculty of Psychology, University of Warsaw. All procedures performed in this study were in accordance with the 1964 Helsinki declaration and its later amendments. All participants gave written informed consent to participate in the study.

## 2.2 Measures

Participants filled out Polish versions of the Edinburgh Handedness Inventory [5] and the Fazio Laterality Inventory [9]. The FLI comprises ten items describing tasks that may be performed using the right or left hand. These are: "Writing," "Drawing," "Waving hello or goodbye," "Using a TV remote," "Snapping your fingers," "Scratching an itchy nose," "Pointing at something in the distance," "Throwing an object," "Reaching to pick up an object," and "Using a hammer." Respondents are asked to indicate the percentage of time (from 0% to 100%) that they use their right hand for these tasks. All items from the original version of the FLI were translated by three psychologists (including the first author) and back-translated to English by a professional translator. The results were compared to the original questionnaire. No major changes were made to the Polish version of the FLI.

The laterality index (LI) from the FLI, which represents hand usage preference, was calculated as per Fazio, et al. [9]: by calculating the average value of the ten items. The LI ranges from 0 to 100: low values indicate a preference for using the left hand, and by analogy, high values indicate a preference for the right-hand. The laterality quotient (LQ) from the EHI was calculated using the formula proposed by Oldfield [5]: subtracting the total answers for the left hand from those for the right hand, dividing the result by the sum of both, and multiplying by 100. The LQ value ranges from −100 to 100. Self-perceived handedness was measured with the use of the FLI. Participants indicated whether they were right-handed, left-handed, or ambidextrous.

## 2.3 Statistical analyses

Data analysis was carried out with RStudio, with a $p$-value of $< 0.05$ considered significant. Rstudio [22] was used to conduct IRT analysis with the EstCRM package [19]. This package

**Table 1. Demographic characteristics of the studied sample.**

| | Female ($n$ = 410) | Male ($n$ = 194) |
|---|---|---|
| Age (SD) | 23.63 (6.43) | 23.99 (6.51) |
| Educational attainment $n$ (%): | | |
| • lower secondary education | 41 (10%) | 24 (12.4%) |
| • high school education | 236 (57.7%) | 119 (61.3%) |
| • non-tertiary education | 6 (1.5%) | 6 (3.1%) |
| • higher education | 126 (30.8%) | 45 (23.2%) |
| Years of education (SD) | 14.94 (2.53) | 14.85 (2.67) |
| Laterality Index from the FLI (SD) | 62.99 (30.84) | 56.39 (31.18) |
| Laterality quotient from the EHI (SD) | 36.28 (74.11) | 22.94 (79.42) |
| Self-perceived handedness (FLI) $n$ (%) | | |
| • Right | 274 (66.8%) | 113 (58.2%) |
| • Left | 111 (27.1%) | 68 (35.1%) |
| • Ambidextrous | 25 (6.1%) | 13 (6.7%) |

includes tools to estimate model parameters for Samejima's continuous response model (CRM) via marginal maximum likelihood estimation and the EM algorithm, to compute item fit residual statistics, to draw empirical 3D item category response curves, and to draw theoretical 3D item category response curves. An IRT model was used to evaluate the psychometric properties of the FLI and to evaluate the performance of each item [23]. Participants with missing data were excluded from the IRT analysis ($n = 123$). In this sample ($n = 604$), the proportion of left-, right- and mixed-handed people was the same as in the entire sample.

To evaluate the fit of one- and two-factor models, we used the lavaan package [24] to conduct confirmatory factor analysis (CFA). This package allows the analysis of both unidimensional and bi-factor models. In contrast to the unidimensional model, the bi-factor model has a general factor that reflects what is common among the items and two or more "group" factors. These additional factors "represent common factors measured by the items that potentially explain item response variance not accounted for by the general factor" [25]. This allows the investigation of a common item variance for all items of the questionnaire and additional common variance for item subsets. CFA was computed with the maximum likelihood (ML) extraction method and with non-orthogonal rotations. Several different indices of goodness of fit were taken into account, including the goodness of fit (GFI), normed fit index (NFI), comparative fit index (CFI), and root mean square error of approximation (RMSEA). All graphs in this paper were prepared using RStudio.

## 3. Results

### 3.1 Reliability

Reliability analysis revealed that the internal consistency of the original scale and its Polish version were nearly the same (Cronbach's alphas of 0.943 and 0.947, respectively). The removal of any items from the scale did not improve Cronbach's alpha score (it did not exceed 0.949).

### 3.2 Concurrent validity

FLI and EHI results were compared in order to check the validity of the FLI. Spearman-rho correlations were calculated for LQ and LI, as well as for parallel items: writing, drawing, and throwing. The correlation between LQ and LI was large, $r_s = 0.757$, $p < 0.001$. This result is higher than that obtained by Fazio, et al. [9], $r_s = 0.517$. We also obtained higher correlations between individual items: writing $r_s = 0.846$, $p < 0.001$ (Fazio's sample: $r_s = 0.562$); drawing $r_s = 0.838$, $p < 0.001$ (Fazio's sample: $r_s = 0.493$) and throwing $r_s = 0.661$, $p < 0.001$ (Fazio's sample: $r_s = 0.533$).

### 3.3 Factor analysis

Confirmatory factor analysis was conducted for comparing the fit of alternative models. Five alternative models were tested: (a) a one-factor model, in which all items loaded on one common factor; (b) a two-factor model, in which items were divided according to the structure proposed by Fazio & Cantor [10]; (c) a second-order factor model, in which handedness was a higher-order dimension that explained the correlation of two primary dimensions; (d) a bi-factor model, in which items were divided into sub-factors as proposed by Fazio & Cantor [10]; and (e) an alternative bi-factor model, in which item 8 was moved to sub-factor 2 (see Table 2).

Both the unidimensional (ML extraction) and two-factor correlated-traits (ML extraction, oblimin rotation) showed very good factor loadings. For the one-factor solution, only items 5 and 6 had factor loadings lower than 0.70 (0.563 and 0.642, respectively). In the two-factor

**Table 2. Results of confirmatory factor analysis for the uni- and multi-dimensional factor models of the FLI.**

| Item | 1 factor model | 2 factor model | | Bi-factor model 1 | | | Bi-factor model 2 | | |
|---|---|---|---|---|---|---|---|---|---|
| | General | Factor 1 | Factor 2 | General | Subfactor 1 | Subfactor 2 | General | Subfactor 1 | Subfactor 2 |
| Writing | 0.924 | 0.984 | | 0.854 | 0.469 | | 0.830 | 0.508 | |
| Drawing | 0.923 | 0.986 | | 0.852 | 0.523 | | 0.825 | 0.566 | |
| Waving | 0.769 | | 0.837 | 0.748 | | 0.347 | 0.760 | | 0.319 |
| TV Remote | 0.806 | | 0.848 | 0.791 | | 0.294 | 0.804 | | 0.263 |
| Snapping | 0.563 | | 0.635 | 0.531 | | 0.387 | 0.535 | | 0.386 |
| Scratching | 0.642 | | 0.750 | 0.593 | | 0.541 | 0.599 | | 0.544 |
| Pointing | 0.772 | | 0.855 | 0.770 | | 0.385 | 0.775 | | 0.374 |
| Throwing | 0.857 | 0.770 | | 0.933 | -0.100[+] | | 0.912 | | 0.022[ns] |
| Reaching | 0.797 | | 0.852 | 0.786 | | 0.331 | 0.795 | | 0.311 |
| Hammer | 0.908 | 0.875 | | 0.889 | 0.197 | | 0.883 | 0.233 | |

All items are significantly related to the factor (or subfactor) at the level of $p < 0.001$ except two which were marked with the symbols:

[+] $p < 0.05$;

[ns]–$p > 0.05$.

model, only item 5 had a factor loading lower than 0.70 (0.635). Thus, factor loadings in both models are satisfactory. Although both models met a final solution, the results of the CFA showed poor fit (see Table 3). The one-factor model reached GFI = 0.636 and RMSEA = 0.251, while the two-factor model had better fit indices (GFI = 0.863 and RMSEA = 0.147), nonetheless suggesting that neither model adequately fit the collected data.

In order to identify a more parsimonious factor structure for the FLI, we conducted second-order and bi-factor analyses. The second-order analysis had weak factor loadings and the model was poorly fitted to the data. Moreover, the estimator for the one factor in this model took on a negative value. Of the multidimensional models, only bi-factor analysis obtained acceptable parameters. In the first step, the sub-factors were determined according to the division proposed by Fazio & Cantor [10]. In this model, the first sub-factor consisted of items 1, 2, 8, and 10. The second sub-factor consisted of the remaining items. Even though the model was found to fit the data adequately (GFI = 0.981; RMSEA = 0.047), item 8 had a negative factor loading value (see Table 2). Since the FLI does not contain any reversed-coded items, this negative value means that item 8 cannot be a part of a first sub-factor. In the second analysis, we decided to add this item to the second sub-factor. The model's fit indices did not change (see Table 3). However, item 8 was found to have a positive but not significant factor loading (0.022; see Fig 1).

**Table 3. Comparison between alternative models of FLI.**

| Model | $\chi^2$ | df | GFI | NFI | CFI | RMSEA 95% CI [LL, UL] | AIC | BIC |
|---|---|---|---|---|---|---|---|---|
| 1 factor model | 1362.238** | 35 | 0.636 | 0.787 | 0.791 | 0.251 [0.24,0.26] | 54336 | 54468 |
| 2 factor model | 480.467** | 34 | 0.863 | 0.925 | 0.930 | 0.147 [0.13,0.16] | 54343 | 54475 |
| Bi-factor model 1 | 58.296** | 25 | 0.981 | 0.991 | 0.995 | 0.047 [0.03,0.07] | 54740 | 54832 |
| Bi-factor model 2 | 65.131** | 25 | 0.981 | 0.991 | 0.995 | 0.047 [0.03,0.07] | 55620 | 55708 |
| Second-order | 276.429** | 33 | 0.925 | 0.957 | 0.962 | 0.111 [0.09,0.12] | 54537 | 54634 |

GFI = goodness of fit; NFI = normed fit index; CFI = comparative fit index; RMSEA = root-mean-square error of approximation; AIC = Akaike information criterion; BIC = Bayesian information criterion;

**$p < 0.001$.

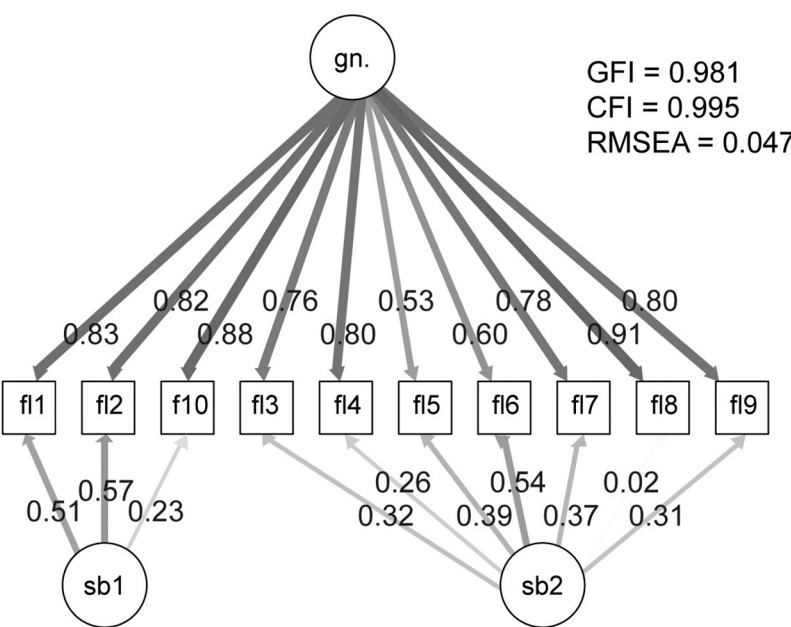

**Fig 1. Bi-factor model of the FLI.** Note: gn. = general factor; fl1–10 = items; sb1 = sub-factor 1; sb2 = sub-factor 2; GFI = goodness of fit; CFI = comparative fit index; RMSEA = root-mean-square error of approximation.

In the last step, the Akaike information criterion (AIC), Bayesian information criterion (BIC), and $\chi^2$ difference were used to compare models (see Table 3). Lower AIC and BIC values indicate better model fit. Within the unidimensional models, the two-factor model was better fit to the data ($\chi^2$ difference = 881.77; $p < 0.001$). Both bi-factor models were better fit to the data than the unidimensional models (415.34; $p < 0.001$). However, the two bi-factor models did not differ from each other.

### 3.4 Item response theory analysis

A continuous response model (CRM) was created to examine the psychometric properties of the tools with responses measured on an interval scale. The results of the CRM are provided in Table 4. In IRT, the discrimination level (a) is considered high if its value is greater than 1.0. Only five items (1, 2, 4, 8, and 10) reached a high discrimination level. The highest

**Table 4. Estimated discrimination and difficulty parameters from the continuous response model.**

| Item | Discrimination Parameter (a) | | Difficulty Parameter (b) | |
|---|---|---|---|---|
| | Estimate | SE | Estimate | SE |
| 1. Writing | 2.249 | 0.065 | -0.327 | 0.019 |
| 2. Drawing | 2.221 | 0.064 | -0.312 | 0.019 |
| 3. Waving | 0.887 | 0.031 | -0.330 | 0.047 |
| 4. TV Remote | 1.016 | 0.033 | -0.358 | 0.041 |
| 5. Snapping | 0.577 | 0.049 | 0.108 | 0.071 |
| 6. Scratching | 0.735 | 0.029 | 0.129 | 0.056 |
| 7. Pointing | 0.917 | 0.031 | -0.223 | 0.045 |
| 8. Throwing | 1.285 | 0.039 | -0.324 | 0.033 |
| 9. Reaching | 0.948 | 0.031 | -0.118 | 0.043 |
| 10. Hammer | 2.000 | 0.058 | -0.384 | 0.022 |

discrimination parameters were reached by item 1 (writing; 2.249) and item 2 (drawing; 2.221). Such high values mean that on these two items most respondents declared using only their dominant hand. A very good discrimination level was reached by item 9 (reaching; 0.948) and item 7 (pointing; 0.917). Item 5 (snapping; 0.577) and item 6 (scratching; 0.735) had the lowest discrimination levels. For these two items, respondents declared the use of both hands regardless of which one was dominant. The difficulty parameter for all items assumed a value oscillating around 0. This optimal value shows that the probability of a right-handed response is the same for all items. However, it should be remembered that the value of this parameter is distorted by the overwhelming number of right-handed people in the group of participants (which corresponds to the values in the general population).

The Item Characteristic Curves (see Fig 2) show the estimated probability of giving an answer (Response Scale) depending on the value of the latent trait Ɵ (Ability Scale). The Response Scale in the Fazio Laterality Inventory ranged from 0 to 100. The Ability Scales ranged from −3 to 3, where 3 corresponds to right-handed people and −3 to left-handed people. The curve for options from 0 to 20 should be high at the lowest ability levels and gradually decline as participants have a higher laterality index. Similarly, the probability of responses from 80 to 100 should be very small at low laterality index levels (Ɵ), but rise as Ɵ increases. The ICCs for items 1, 2, 8, and 10 had a different pattern. They indicated that, for almost all items, the individuals responded in a very dichotomous manner. The low variability of the results on the Ability Scale is a result of such unambiguously extreme responses, without answers from the middle of the scale. This effect occurs when the distribution for the variable is negatively skewed with reduced variability [17]. It is worth mentioning that the curves present a theoretical model in which the probability of response from the middle scale is estimated. Such results indicate that these four items are strongly associated with the laterality index and are good at differentiating hand preference without using answers from the middle of the scale.

The discrimination parameters and the category response curves for the items with lower discrimination parameters (3, 5, 6, and 7) indicated that activities from these items were done with both hands, with a slight predominance of the dominant hand. For example, most right-handers scratched their nose with their right hand. However, for values from 90 to 100 (on the Response Scale), the probability level decreased. Such a result is not surprising because many right-handers sometimes scratch their nose with their left hand. Thus, the probability that they scratch exclusively with their right hand is low.

Due to the fact that the monotonicity criterion was not met for several variables, we checked if there were any anomalies that may have arisen due to violation of this assumption. None of the items had low discrimination parameters. However, the theoretical distribution of curves does not correspond to dichotomous data (for some variables), so it was decided to draw Empirical Item Characteristic Curves (see Fig 3). Here the Ability Scale ranges from −1.5 (for left-handers) to 1 (for right-handers). The curves for items 1, 2, and 10 clearly showed that the vast majority of right-handed people declared using their right hand to perform these activities. For items 5 and 6, it was clearly visible that participants used their dominant hand for snapping and scratching an itchy nose only for 60% of the time—regardless of which hand was dominant.

The items exhibited a narrow range of difficulty parameters. Since the majority of participants were right-handed (63.9% of the sample analyzed in the IRT model), most of them answered most questions correctly—that is, according to the key. Only items 5 and 6 had higher difficulty parameters (0.108 and 0.129, respectively). For these activities (snapping and scratching), fewer people used their dominant hand exclusively.

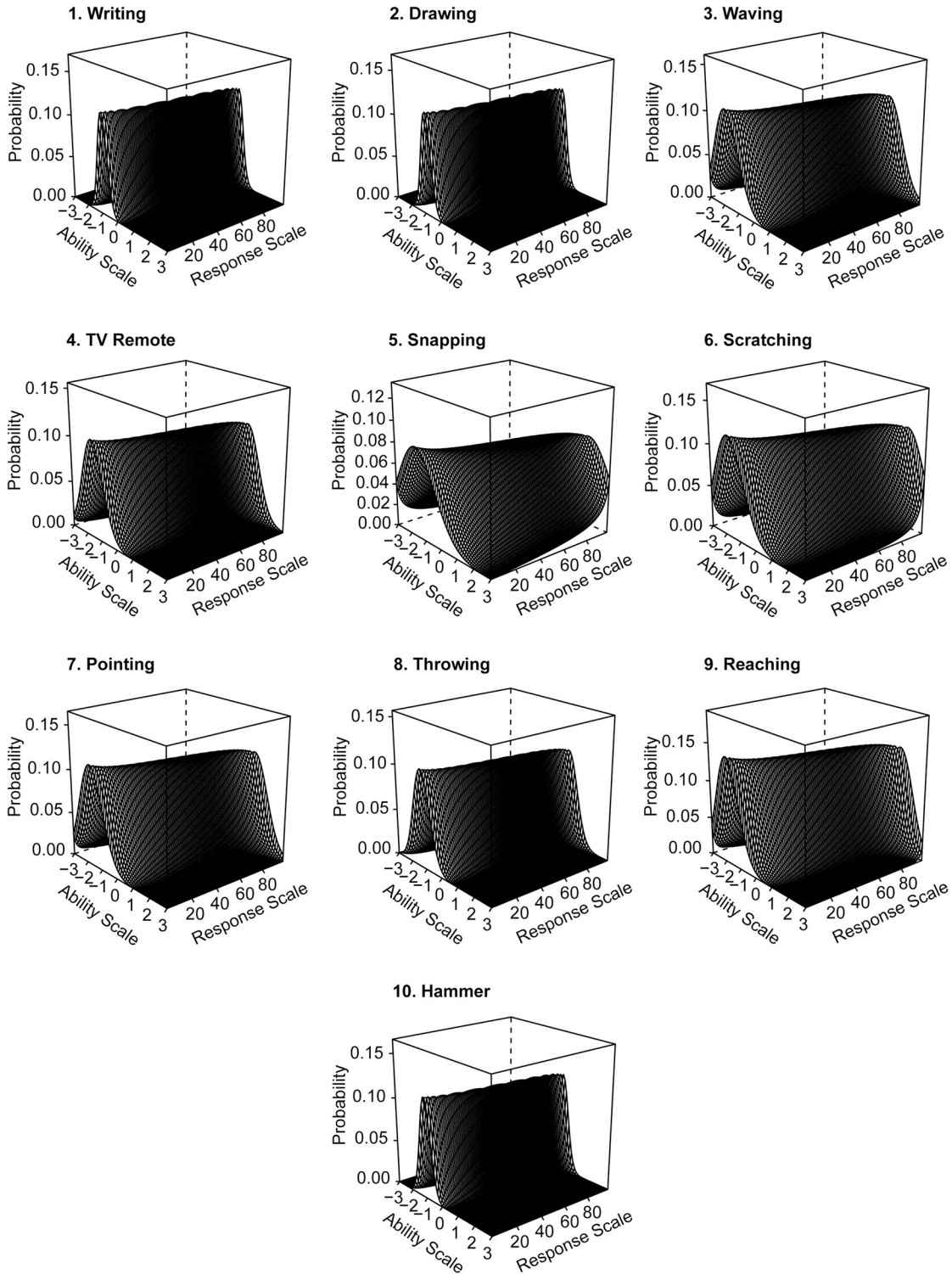

**Fig 2. Theoretical three-dimensional Item Characteristic Curves (ICCs) for FLI items.** Note: the functions describe the probability that a person with greater lateralization index value (Ability Scale) will provide a response on the FLI scale (Response Scale) which ranges from 0 to 100 points. Lower levels on the Ability Scale represent left-handed people, higher values represent right-handed people.

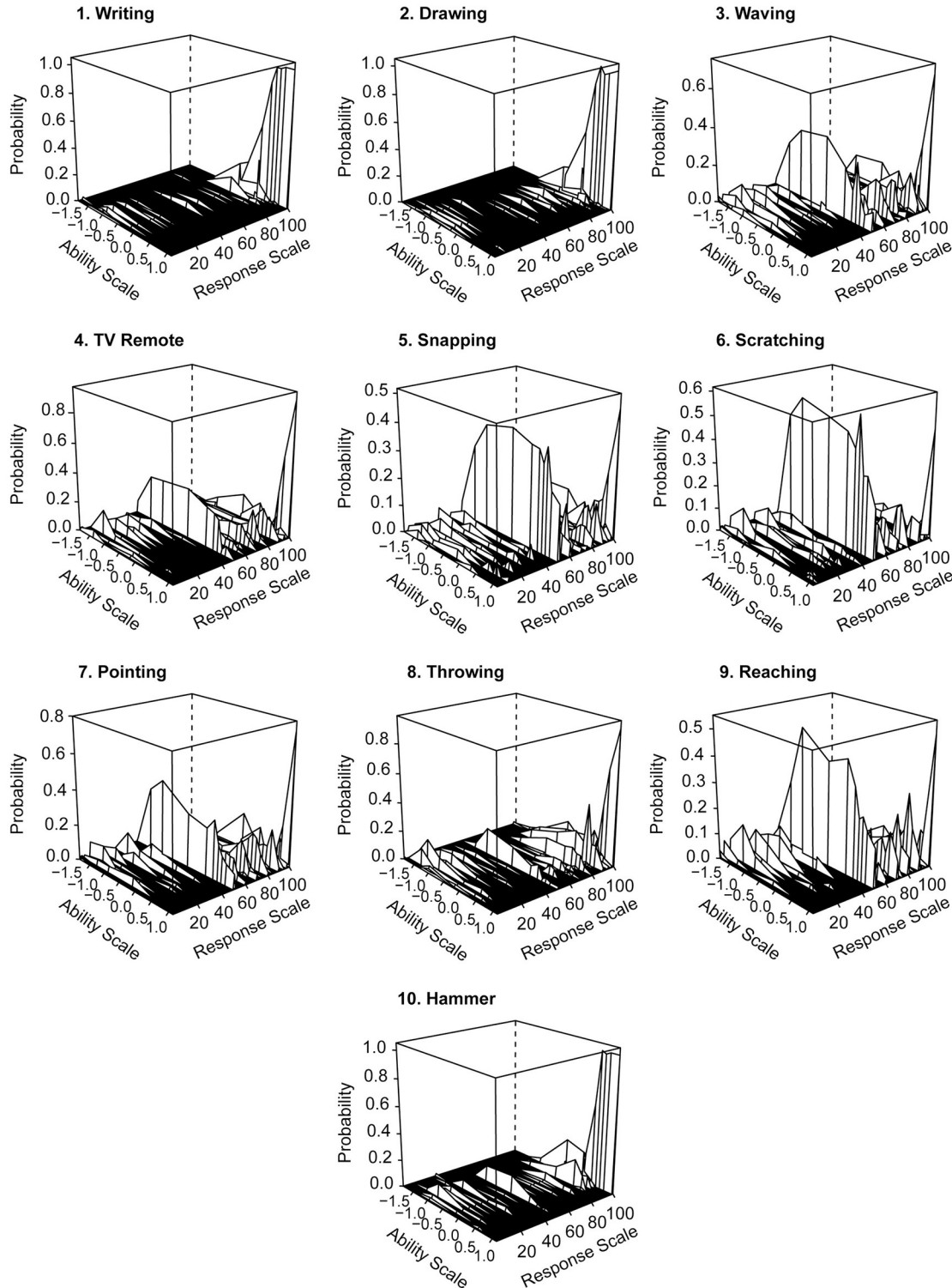

**Fig 3. Empirical three-dimensional Item Characteristic Curves for FLI items.** Note: The functions show with what probability a person with a given laterality index (Ability Score) chooses the answer on the FLI scale. Lower values on the Ability Scale represent left-handed people, higher values represent right-handed people.

Item 4 (using a TV remote) had the lowest difficulty estimate in the empirical model (−0.534). This indicates that even left-handed people report that they often use their right hand to change the TV channel. Conversely, item 5 (snapping) and item 6 (scratching an itchy nose) had the highest difficulty estimates in the empirical model, at −0.116 and −0.123, respectively. This means that left-handed people rarely used their right hand for these activities.

## 4. Discussion

Laterality measurement is essential for designing and performing experiments which use neuroimaging methods. Additionally, recent data (e.g., [26]) has shown the usefulness of measuring laterality for identifying biological bases of sexual orientation. An example of the recent improvements in measuring laterality is the Fazio Laterality Inventory (FLI). In a series of studies, Fazio, et al. [9, 10] demonstrated that this questionnaire has good psychometric properties. The aim of our study was to establish the basic psychometric properties of the FLI using an independent sample. We also wanted to expand knowledge of the validity of this measure by applying IRT models.

The reliability of the Polish version of the Fazio Laterality Inventory was very high and comparable to the one estimated for the original version of the questionnaire. We tested the concurrent validity of the FLI by performing a correlation analysis with the Edinburgh Handedness Inventory. Interestingly, we found higher correlation coefficients than Fazio, et al. [9], both for laterality indices and individual items that measure the same skill. This is probably due to the larger sample size in our study. An increase in sample size may both affect the precision of the measurement of correlated variables and the estimation of correlation in the population. Together, these may impact the correlation size [27].

We validated the Fazio Laterality Inventory by examining its factor structure. Although the exploratory analysis suggested a one-factor structure, the confirmatory analysis did not corroborate this finding. Next, we tested a two-factor model, as proposed by Fazio & Cantor [10], and a second-order model. Both did not fit the data adequately. In the next step, we evaluated two versions of the bi-factor model. This model assumes that a general latent factor can explain the covariance among all observed variables and that additional common variance among groups of indicators can be accounted for by one or more latent group factors [13]. In recent years, many studies have applied bi-factor models to identify the structure of psychopathology [28], personality [29], and intelligence [30]. More importantly, Musálek [12] identified a bi-factor structure for his measure of laterality for children. Our analysis revealed that both bi-factor models fit the data most appropriately. In a sample from the general population, the analysis of Fazio, et al. [9] produced a one-factor solution, while Fazio & Cantor [10] identified a two-factor structure in a sample with established atypical handedness. The sample in our study was also taken from the general population. In such a case, Fazio & Cantor [10] anticipated that a one-factor solution for the FLI would be produced. As stated by Reise, et al. [25], the bi-factor model is an excellent framework for studying how heterogeneous item content may be applied to measure unidimensional constructs. IRT analysis provided proof that the FLI items constitute two different sets of indicators of laterality. The first one consists of four items (1, 2, 8, and 10) that are strongly associated with the laterality index. This set is identical to the first factor identified by Fazio & Cantor [10] as "fine motor/ballistic movements." The second set encompasses the rest of the items. Together they constitute the second factor, which was named "expressive/instrumental movements." The general factor represents the broad construct which the scale is intended to measure (i.e., handedness).

It should be noted that in the bi-factor models identified in factor analysis, item 8 (Throwing) was loosely related to both sets of items. The lack of confirmation of the results of the

study of Fazio, et al. [9] in our sample may be interpreted as a sign of the instability of the factor structure of the FLI. This could also be interpreted as cultural differences being the cause of differences in the factor structure, as cultural factors have been implicated in differences in handedness [31]. Recently Espírito-Santo, Pires, Garcia, Daniel, Silva da & Fazio [32] suggested that different factorial solutions of the Edinburgh Handedness Inventory may result from cultural differences, primarily due to different interpretations of specific items [33]. As the Fazio Laterality Inventory has only been validated in North American populations, our suppositions about the nature of the divergence from expectations of the factor structure of the questionnaire need to be further tested with other samples.

Regarding the margin of the bi-factor solutions detected in our sample, it is worth mentioning that in such cases, a general factor may be seen as a proxy of methodological issues, particularly as an index of method variance [34]. When discussing the results of their study, Fazio & Cantor [10] argue that the dissociation of the factors they obtained stems from low method variance. Assuming that the bi-factor structure of the FLI reflects methodological issues with this questionnaire, the results of our study contradict Fazio & Cantor's conclusions. In their analysis of sources of method bias, Podsakoff, MacKenzie & Podsakoff [35] pointed to the potential influence of different response styles. In our analysis, the category response curves for items 1, 2, 8, and 10 revealed that participants responded to these four items in a very dichotomous way. This may suggest that the response format, probably together with the content of these items, facilitates the extreme response style (ERS). The seminal paper of Baumgartner & Steenkamp [36] indicated how the ERS may promote method bias while Arce-Ferrer & Ketterer [37] showed its impact on factor structure. It should be noted that some authors [38] have suggested that the use of a 101-point scale (as in the FLI) may enhance respondents' ambiguity, which is the basis of the ERS [39]. Furthermore, cultural factors are also seen as a potential source of the ERS [40]. In terms of Hofstede's [41] dimensions, Polish society scores higher than American society on uncertainty avoidance (a construct similar to tolerance of ambiguity). Therefore, we speculate that the interaction of the current response format of the FLI and the cultural norms of Polish respondents may produce a specific response style [42]. Of course, this supposition should be further tested. Another response style which may promote method bias is rounding [43]. In an experiment testing different versions of 101-point scales, Liu & Conrad [44] found that an open-ended format resulted in more frequent rounding. A quick examination of the frequency of answers to items on the Polish version of the FLI revealed that most of them are rounded and divisible by 10. This is likely to be another factor that influenced method bias in our sample. To sum up, we suggest that the specific response format of the FLI may be a factor influencing fluctuations in method variance and thereby also the factor structure of the questionnaire. However, it should be noted that Fazio & Cantor's [10] study was conducted using a sample from a population that is neurologically atypical, while we used a sample from the general population. Thus, all the differences in the factor structure observed by us may result from differences in the phenotype of the studied samples. Such a conclusion stresses the need for future studies on the factor structure of the FLI using both neurologically typical and atypical populations.

When discussing the FLI's properties, it is worth noting that all items have a difficulty parameter close to 0. This means that the probability of giving an answer indicating being right-handed is about 50%, suggesting that the activities selected for the questionnaire are very good. The slightly negative values of the difficulty parameter are due to the prevalence of right-handed people in the group of respondents. The adequacy of the selection of activities for the FLI is illustrated by item 8: as mentioned above, this item is loosely related to the others and was answered in a dichotomous way in our sample. However, the IRT analyses showed that it has excellent discrimination and difficulty parameters, making it a strong candidate for

measuring the laterality construct. It is worth emphasizing, however, that despite the original assumption that laterality can be captured on a continuous and not a dichotomous scale, the four items (from the "fine motor/ballistic movement" sub-factor) were actually very dichotomous.

Our study has some important strengths. It is the first to use the IRT approach to identify the psychometric properties of the FLI [9] and does so in a relatively large sample. Nevertheless, it is not free from limitations. As in previous studies on the FLI [9, 10], our sample was not representative. There were more left-handed people in our sample than the population average, which is likely because they were disproportionately attracted to a study about handedness. The majority of the participants were women, which could have influenced the results: men are generally more likely to be left-handed [45], probably as a result of sex differences in brain development. Moreover, Fazio, et al. [9] found gender differences in the laterality index —men were less likely to give extreme responses. However, it should be noted that the ratio of men to women in our sample was very similar to that of Fazio, et al. [9] (65.3% women in their aggregate sample vs. 67.9% in our sample), which provides a good basis to make comparisons between the studies. Lastly, gender identity and sexual orientation were not controlled for in our analyses, and there are indications that these are also implicated in laterality [26, 46].

## 5. Conclusion

Our analyses depict the Fazio Laterality Inventory as a highly reliable tool with a bi-factor structure. The items have excellent discrimination and difficulty parameters, indicating that the activities it describes are well suited to measure handedness. Thus, our study adds to the evidence that it is a high-quality tool and researchers should consider it a better alternative to the still commonly used Edinburgh Handedness Inventory.

## Supporting information

**S1 File.**
(DAT)

## Acknowledgments

The authors would like to thank Agata Grysiak and Magdalena Bochyńska for help with data collection.

## Author Contributions

**Conceptualization:** Wojciech Łukasz Dragan.

**Data curation:** Wojciech Łukasz Dragan, Andrzej Śliwerski, Monika Folkierska-Żukowska.

**Formal analysis:** Andrzej Śliwerski.

**Funding acquisition:** Wojciech Łukasz Dragan.

**Investigation:** Wojciech Łukasz Dragan, Monika Folkierska-Żukowska.

**Methodology:** Wojciech Łukasz Dragan, Andrzej Śliwerski.

**Project administration:** Monika Folkierska-Żukowska.

**Software:** Andrzej Śliwerski.

**Visualization:** Andrzej Śliwerski.

**Writing – original draft:** Wojciech Łukasz Dragan, Andrzej Śliwerski, Monika Folkierska-Żukowska.

**Writing – review & editing:** Wojciech Łukasz Dragan, Andrzej Śliwerski, Monika Folkierska-Żukowska.

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
