## [Decision Letter · Decision Letter 0]

15 Dec 2021

PONE-D-21-31396New Data on the Validity of the Fazio Laterality InventoryPLOS ONE

Dear Dr. Dragan,

Thank you for submitting your manuscript to PLOS ONE. After careful consideration, we feel that it has merit but does not fully meet PLOS ONE’s publication criteria as it currently stands. Therefore, we invite you to submit a revised version of the manuscript that addresses the points raised during the review process.

We look forward to receiving your revised manuscript.

Kind regards,

Frantisek Sudzina

Academic Editor

PLOS ONE

Journal Requirements:

Reviewers' comments:

Reviewer's Responses to Questions

**Comments to the Author**

1. Is the manuscript technically sound, and do the data support the conclusions?

Reviewer #1: Yes

2. Has the statistical analysis been performed appropriately and rigorously? 

Reviewer #1: Yes

3. Have the authors made all data underlying the findings in their manuscript fully available?

Reviewer #1: Yes

4. Is the manuscript presented in an intelligible fashion and written in standard English?

Reviewer #1: Yes

5. Review Comments to the Author

Reviewer #1: This is a well-conducted additional validation of the FLI. Strengths include the relatively large sample size and the variety of analyses performed.

Minor points to address in the manuscript are as follows:

1. Formatting/stylistic issues. On page 3, like 62, the author's first name is included, and this appears to be the only place in the manuscript formatted in this manner. Page 5 line 95 has the citation formatted incorrectly. On page 7, several quotation marks are ostensibly backwards from how they are typically used. Citation brackets also appear to be missing on page 9, line 174. Formatting of citation, page 19, line 300. There are a few other instances of the citation brackets being missing as well.

2. Although this is a proof at this point, it would have been nice for Table 2 to fit on one page. Obviously this will need reduced going forward. It will be more effective to see all the factor loadings with one glance.

3. Page 20, line 346 - the use of "American" here needs clarified. It is assumed the authors do not mean the USA, given that the FLI has been validated in the USA and Canada.

4. The authors seem to state in the discussion that the different results found in their validation of the FLI have to do with method variance. Although they state that further testing is necessary, it seems as though their phrasing downplays the potential differences found between neurologically normal populations and likely neurologically atypical populations. It may be worth discussing both sides of this issue further, since this is potentially a major issue given the populations in which neuroimaging studies are usually conducted.

Relatively larger issues which need to be addressed include:

1. It was not made clear in the manuscript if the participants were asked their self-perceived handedness. If so, this should be explicitly stated and also these results included in Table 1. If not, this methodological choice should be addressed.

2. Related to the above, is there data available on the typical rates of right and left handedness in a Polish population generally? The EHI LQs in table 1 seem very low, and this raises the question of the cultural acceptance of sinistrality or other cultural factors in these results.

3. It seems the authors may need to comment on the amount of missing data as presented on page 8 in regards to why this may have happened.

4. On page 15, it is mentioned that there is an "overwhelming number of right-handed people in the group of participants." Although this is surely true from the FLI and EHI values, what is the basis for this? Is it based on self-report or cut scores for the instruments? If so, what cuts were used? How would participants be classified with one vs. the other instrument? This would all be useful information to present.

6. PLOS authors have the option to publish the peer review history of their article (what does this mean?). If published, this will include your full peer review and any attached files.

Reviewer #1: No

---

## [Author Response · Author response to Decision Letter 0]

4 Jan 2022

Responses to the reviewer's comments

We want to thank the Reviewer for taking the time to review the manuscript. We sincerely appreciate all their valuable comments and suggestions, which helped us improve the manuscript's quality.

1. Formatting/stylistic issues. On page 3, like 62, the author's first name is included, and this appears to be the only place in the manuscript formatted in this manner. Page 5 line 95 has the citation formatted incorrectly. On page 7, several quotation marks are ostensibly backwards from how they are typically used. Citation brackets also appear to be missing on page 9, line 174. Formatting of citation, page 19, line 300. There are a few other instances of the citation brackets being missing as well.

Thank you for this valuable comment. We included all the suggested corrections.

2. Although this is a proof at this point, it would have been nice for Table 2 to fit on one page. Obviously, this will need reduced going forward. It will be more effective to see all the factor loadings with one glance.

We changed the format of Table 2 according to the Reviewer’s suggestion.

3. Page 20, line 346 - the use of "American" here needs clarified. It is assumed the authors do not mean the USA, given that the FLI has been validated in the USA and Canada.

We clarified this statement by adding the "North" to "American."

4. The authors seem to state in the discussion that the different results found in their validation of the FLI have to do with method variance. Although they state that further testing is necessary, it seems as though their phrasing downplays the potential differences found between neurologically normal populations and likely neurologically atypical populations. It may be worth discussing both sides of this issue further, since this is potentially a major issue given the populations in which neuroimaging studies are usually conducted.

We changed the discussion according to the Reviewer’s suggestion.

5. It was not made clear in the manuscript if the participants were asked their self-perceived handedness. If so, this should be explicitly stated and also these results included in Table 1. If not, this methodological choice should be addressed.

Accidentally we omitted this information in the previous version of the manuscript. However, it was included in Table 1. In addition, a short explanation of how this data was gathered was added in the Method section.

6. Related to the above, is there data available on the typical rates of right and left handedness in a Polish population generally? The EHI LQs in table 1 seem very low, and this raises the question of the cultural acceptance of sinistrality or other cultural factors in these results.

Unfortunately, there is no available normative data on handedness in a general Polish population. 

7. It seems the authors may need to comment on the amount of missing data as presented on page 8 in regards to why this may have happened.

Such an amount of missing data was probably a consequence of the online format of the survey (no force response has been applied). Since the same data was used to estimate the factor structure and IRT analysis, we decided to remove all cases with at least one non-response (the software used for the IRT required no missing data). Due to the technical nature of this explanation, we decided not to include it in the discussion.

8. On page 15, it is mentioned that there is an "overwhelming number of right-handed people in the group of participants." Although this is surely true from the FLI and EHI values, what is the basis for this? Is it based on self-report or cut scores for the instruments? If so, what cuts were used? How would participants be classified with one vs. the other instrument? This would all be useful information to present.

This observation was based on the self-report, which, as mentioned above, was added to Table 1.

---

## [Editor Report · Decision Letter 1]

6 Jan 2022

New Data on the Validity of the Fazio Laterality Inventory

PONE-D-21-31396R1

Dear Dr. Dragan,

We’re pleased to inform you that your manuscript has been judged scientifically suitable for publication and will be formally accepted for publication once it meets all outstanding technical requirements.

Kind regards,

Frantisek Sudzina

Academic Editor

PLOS ONE
---

## [Editor Report · Acceptance letter]

7 Jan 2022

PONE-D-21-31396R1 

New Data on the Validity of the Fazio Laterality Inventory 

Dear Dr. Dragan:

I'm pleased to inform you that your manuscript has been deemed suitable for publication in PLOS ONE. Congratulations! Your manuscript is now with our production department. 

Kind regards, 

on behalf of

Dr. Frantisek Sudzina 

Academic Editor

PLOS ONE